# A study on the characteristics of coke in the hearth of a superlarge blast furnace

Qing Q. Lv[1,2,3], Yong S. Tian[1,2], Ping Du[4], Jun L. Zhou[4], Guang H. Wang[1,2]*

**1** School of Chemistry and Chemical Engineering, Wuhan University of Science and Technology, Wuhan, China, **2** Key Laboratory of Hubei Province for Coal Conversion and New Carbon Materials, Wuhan, China, **3** Technology Research and Development Department, Jiangsu Huade Hydrogen Energy Technology Co., Ltd., Zhangjiagang, China, **4** Iron making and Environment Research Office, Iron and Steel Research Institute of Shasteel, Zhangjiagang, China

* wghwang@263.net

**Data Availability Statement:** All relevant data are within the paper.

**Funding:** This work was only financially Supported by National Key R&D Program of China (2017YFB0304300&2017YFB0304303). The

## Abstract

An in-depth study on the characteristics of coke in the hearths of blast furnaces is of great significance for explaining the mechanism of coke deterioration in blast furnaces. In the present work, the changes in macromorphology, degree of graphitization, and microstructure of the coke taken from different hearth locations of a 5,800 m³ superlarge blast furnace during its intermediate repair period were systematically studied. Significant differences were found between cokes obtained from the edge ("edge coke") and from the center ("center coke") of the hearth in terms of properties and degradation mechanisms. Edge coke was severely eroded by liquid metal, and only a small amount of slag was detected in the coke porosity, whereas center coke was basically free from erosion by liquid metal, and a large amount of slag was detected in the coke porosity. The degree of graphitization of edge coke was higher than that of center coke. The carburizing effect of liquid metal was the main cause of the degradation of edge coke and made it smaller or even disappear. Center coke was degraded due to the combination of two factors: slag inserted into micropores on the surface of center coke loosened the surface structure; and graphite-like flakes that appeared on the center coke surface lowered the strength and caused cracks in the surface.

## 1. Introduction

Coke serves as the structural support, fuel, reducing agent, and carburizer in blast furnaces [1, 2]. As iron-making technologies advance, blast furnaces tend to be larger with higher rates of pulverized coal injection (PCI), a trend that raises higher requirements for coke quality [3]. As a response, exploring the mechanism of coke deterioration in blast furnaces is important for developing iron-making technologies. In the blast furnace process, high temperature and high pressure with extremely hostile conditions make it impossible to monitor the reactions in the furnace involving coke. This circumstance has greatly limited researchers' understanding of the deterioration mechanism of coke in blast furnaces. Huiqing Tang [4], Zhen Miao [5] and Yuting Zhou [6] established a mathematical

funder is National Key R&D Program of China. And no sponsors of funder play any role in the study design, data dollection and analysis, desision to publish, or preparation of the manuscript. The National Key R&D program provided support in the form of salaries for authors [QQL, YST, PD, JLZ and GHW], but did not have any additional role in the study design, data collection and analysis, decision to publish, or preparation of the manuscript. The specific roles of these authors are articulated in the 'author contributions' section.

simulation model to study the combustion behavior of coal and coke in the blast furnace tuyere zone. Gerd Rantitsch [7] found that the thermochemical reactivity and strength of blast furnace coke at 1100˚C depend on the state of graphitization of the feed coke. Samples taken from the tuyere area of the blast furnace show that carbonaceous matter graphitization occurs at temperatures above 1900˚C. Zhang's [8] research shows that the aluminosilicate in coke hardly changes in physical properties in the process of coke gasification in a blast furnace, but it combines with alkali materials. The iron-bearing phase has a high activity and strong interaction with the carbon matrix. Iron was proven to have a strong catalytic effect on graphitization in the temperature range of 1100–1500˚C. [9–11] With the tuyere sampling technique, researchers have obtained samples from operating furnaces, performed basic studies on the structure [12–15], mineral transformation [16–20], reactivity, and postreaction strength of coke [21, 22], and improved the understanding of the process by which coke deteriorates in the lower bottom part of a furnace. Zhiyu Chang's research shows that the pores in the coke of the cohesive zone of a blast furnace are filled with slag. The main composition of slag is chlorite or anorthite. Spinel is also found in the cracks of coke. The stress produced by spinel crystal growth may lead to coke cracking.[23] Zhihao Zeng [24] took samples from the inside of a blast furnace during maintenance. The effect of alkali metals on coke catalysis in a blast furnace was studied. Studies show that the content of alkali metals in tuyere coke is the most important factor affecting gasification reactivity, and the secondary factors are the carbon chemical structure and pore structure. However, the sample amount, sampling location, and sampling depth are all restricted because a sampling tube must be inserted into the blast furnace, and it is impossible to obtain all representative samples from the feedstock in the lower bottom part of the hearth. For the 5,800 m$^3$ superlarge blast furnace considered in the present work, the central part of the deadman cannot be reached using tuyere sampling. In addition, tuyere sampling is also limited to the horizontal direction at the tuyere level and cannot reach the coke below it, which prevents researchers from learning about the deterioration mechanism of the coke below the tuyere level.

The 5,800 m$^3$ superlarge blast furnace of the Shagang Group is one of the largest blast furnaces in the world. Coke samples were obtained from different positions during blast furnace intermediate repair. Cokes were taken from different heights from the edge of the blast furnace to the center of the deadman. The cokes in these locations are not available for tuyere sampling technology. Instruments including a polarizing microscope, X-ray diffraction (XRD) analyzer and a scanning electron microscopy (SEM) system were used to systematically characterize the particle size, macromorphology, and microstructure of the coke in an effort to explore the deterioration mechanism of coke in the hearth of a superlarge blast furnace.

## 2. Sampling locations and analytical methods

### 2.1 Sampling locations

The diameters on the tuyere level of the blast furnace are 15 meters, and there are 40 tuyeres and 3 tap holes for tapping iron around the clock. The blast furnace was put into operation in Oct. 2009 and operated smoothly at approximately 12,500 t HM/d. The coke and coal consumption rates were maintained at an average of 380 kg/t HM and 152 kg/t HM, respectively. The average M40 (cracking resistance index) and M10 (abrasive resistance index) of the coke charged into the blast furnace were 90 and 5.7, respectively. After the blow-out and cool-down of the furnace, the hearth was dissected, and coke samples were collected manually. The sampling location is presented in Fig 1. Eight coke samples were obtained at locations between 0 and 7.5 meters inward from the hearth edge 1 meter below the tuyere level and at locations

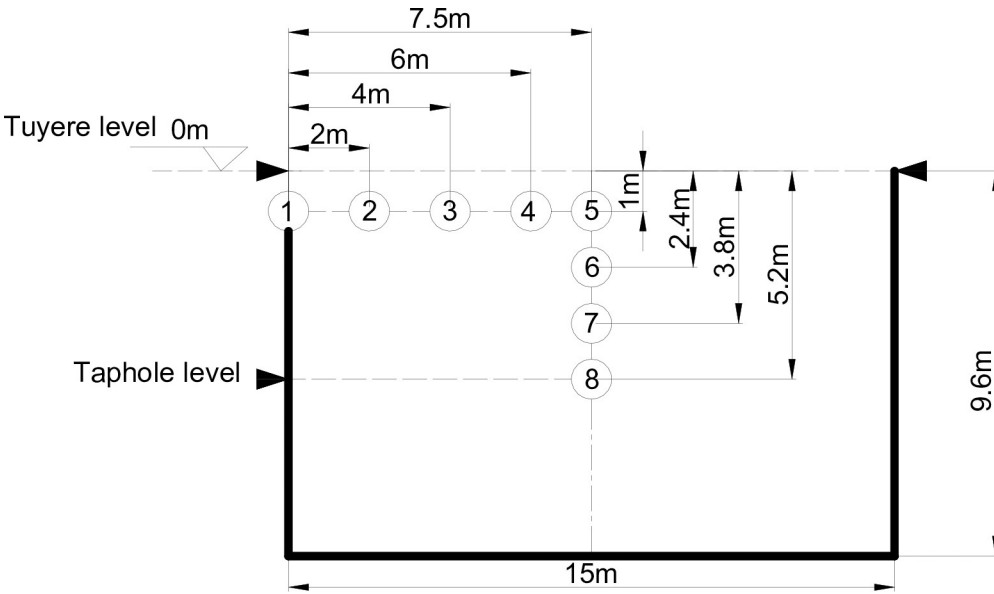

**Fig 1. Sampling locations.**

between 1 and 5.2 meters downward from the tuyere level on the centerline of the furnace hearth.

## 2.2 Analytical methods

First, coke was used to observe the apparent morphology, and then a 2 kg coke sample was ground into a powder whose particle size was lower than 0.075 mm. Out of the powder, a sample of 100 g was taken and observed under an XRD analyzer (SHIMADZU XRD-6100) to analyze the changes in the microcrystalline texture.

Grade 25–40 mm coke was chosen for analysis. First, foreign matter attached to the coke was removed, and then the coke was ground into slices of 20 mm × 20 mm × 2 mm (thickness) by a grinding-polishing machine. The microstructures of the slices were observed with their energy spectra analyzed by SEM coupled with EDS. Subsequently, the slices were observed under an optical microscope to characterize the microstructures of the pore walls and optical properties of the substrates.

Under the optical microscope, the coke slices are divided into several grids, and the microstructure in the grids is counted. The porosity of unfilled slag is 0, that of filled slag is 1, and that of the coke structure is 2. The meshing method is shown in Fig 2. The slag area to pore area ratio is calculated according to the following equation.

The slag area to pore area ratio = number of slags detected/ (number of slags detected + number of pores detected) × 100%

## 3. Results and discussions

### 3.1 Macromorphologies of the coke from different locations

As shown in Fig 3A–3C, the erosion marks of liquid metal left on the coke surfaces are clearly identified on the edge coke at 0 m, 2 m and 4 m from the hearth edge, with flake-like structures observed; its macromorphologies are presented in Fig 3I, from which clear erosion marks are apparent. According to Fig 3D and 3(e), the coke at 6 m and 7.5 m from the

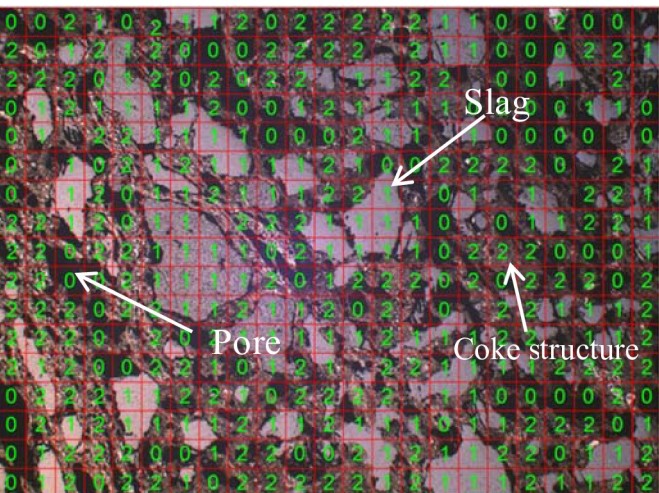

**Fig 2. Meshing method.**

hearth edge has its bulk shape remaining, with clear edges and angles apparent. Fig 3F–3H demonstrate that as the locations go deeper into the bottom of the hearth, the bulk shape of the coke does not undergo substantial changes. It is observed from Fig 3J and 3K that residual slag and iron at 0 m and 2 m from the hearth edge form bulks of 5–10 cm in size, whereas slag and iron at 6 m and 7.5 m form liquid drops of 1–2 cm in size. This result implies that the area between 5.5 m and 7.5 from the hearth edge is the deadman of the 5,800 m$^3$ blast furnace. Because the hearth diameter is as large as 15 meters, which makes it impossible for the air blast from tuyeres to blow through the center of the deadman and results in poor air permeability and poor fluid permeability, the inflow of liquid metal and consumption of coke in the deadman are greatly reduced, leaving the macromorphology of the coke in the deadman basically unchanged with clear edges and angles being apparent. The low air permeability and low fluid permeability in the deadman cause the liquid metal to primarily distribute along the hearth edge of blast furnaces. Coke is rapidly consumed by liquid metal by the carburizing mechanism. This process assists the formation of groove marks on the coke surfaces. The distribution of liquid metal in the hearth is shown in Fig 4, where it is seen that the amount of center slag and iron is less than the amount of edge slag and iron.

## 3.2 Microstructures of the coke from different locations

It is observed from Fig 5A–5G that the coke between 0–7.5 m from the hearth edge shows dark red microstructures under the polarizing microscope with an oil immersion lens (×500 magnification), a kind of isotropic structure similar to that of superpure graphite, as shown in Fig 5(1). The microstructure of the original coke under the microscope (×500 magnification) has an anisotropic, coarse mosaic texture, as shown in Fig 5K. Reports [25, 26] conclude that high temperature drives the transformation from a coke structure to a graphite structure. This finding indicates that the coke in the hearth of the blast furnace has been graphitized because of the prolonged high temperature. According to Fig 5A, 5B and 5I, the pore walls of the coke have turned coarse and thin, indicating that the coke has undergone fierce gasification. This result is due to the closeness of the coke to tuyeres, through which a large amount of oxygen is injected into the furnace, gasifies the coke fiercely, and leads to erosion of the coke. According to Fig 5B–5J, semitranslucent slag is

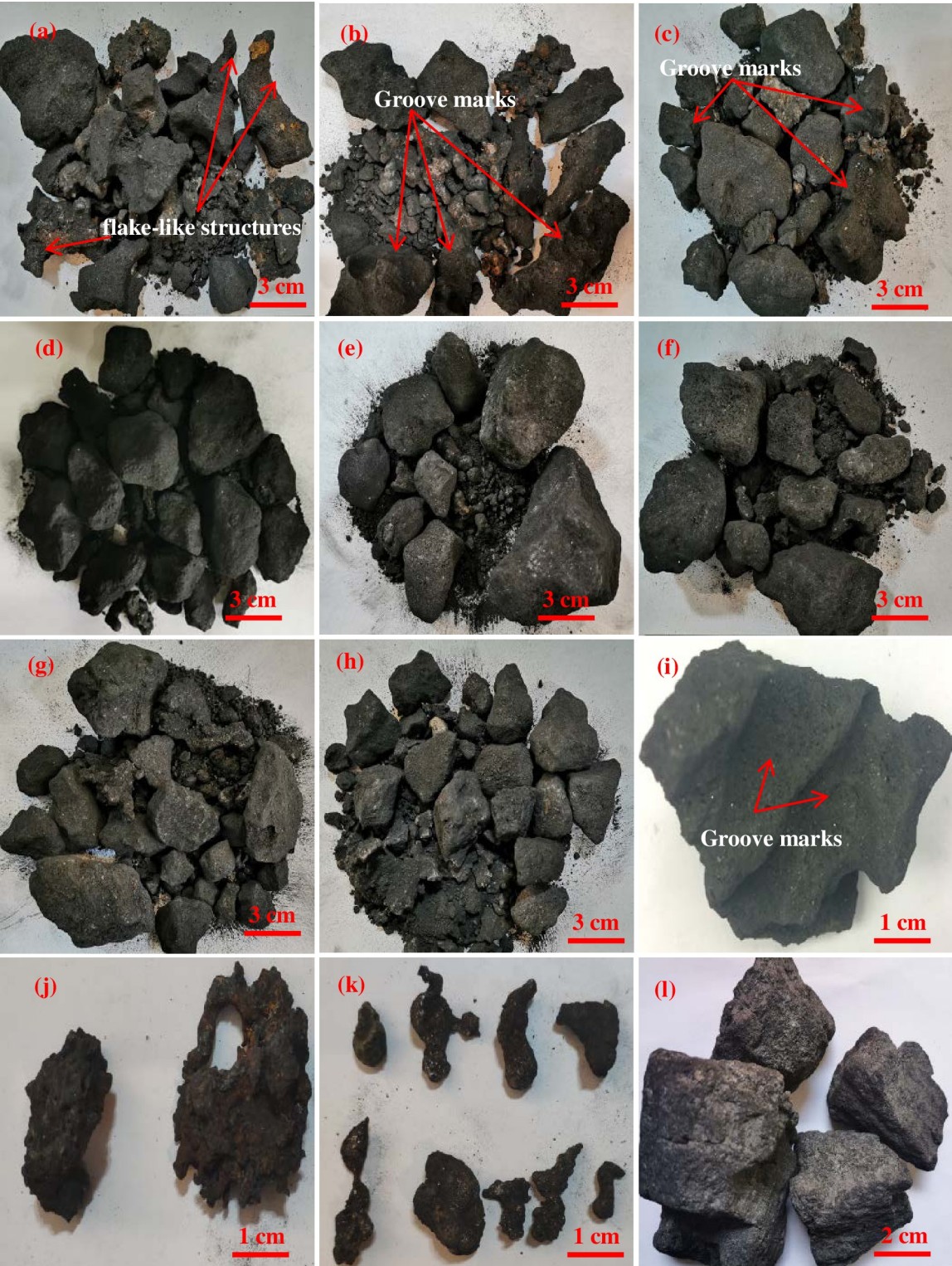

**Fig 3. Macromorphologies of the coke in the hearth.** (a) Sample ① at 0 m horizontal and 1 m vertical; (b) Sample ② at 2 m horizontal and 1 m vertical; (c) Sample ③ at 4 m horizontal and 1 m vertical; (d) Sample ④ at 6 m horizontal and 1 m vertical; (e) Sample ⑤ at 7.5 m horizontal and 1 m vertical; (f) Sample ⑥ at 7.5 m horizontal and 2.4 m vertical; (g) Sample ⑦ at 7.5 m horizontal and 3.8 m vertical; (h) Sample ⑧ at 7.5 m horizontal and 5.2 m vertical; (i) coke eroded by liquid metal at 0–2 m horizontal and 1 m vertical; (j) slag and iron at 0–4 m horizontal and 1 m vertical; (k) slag and iron at 6–7.5 m horizontal and 1 m vertical; and (l) original coke.

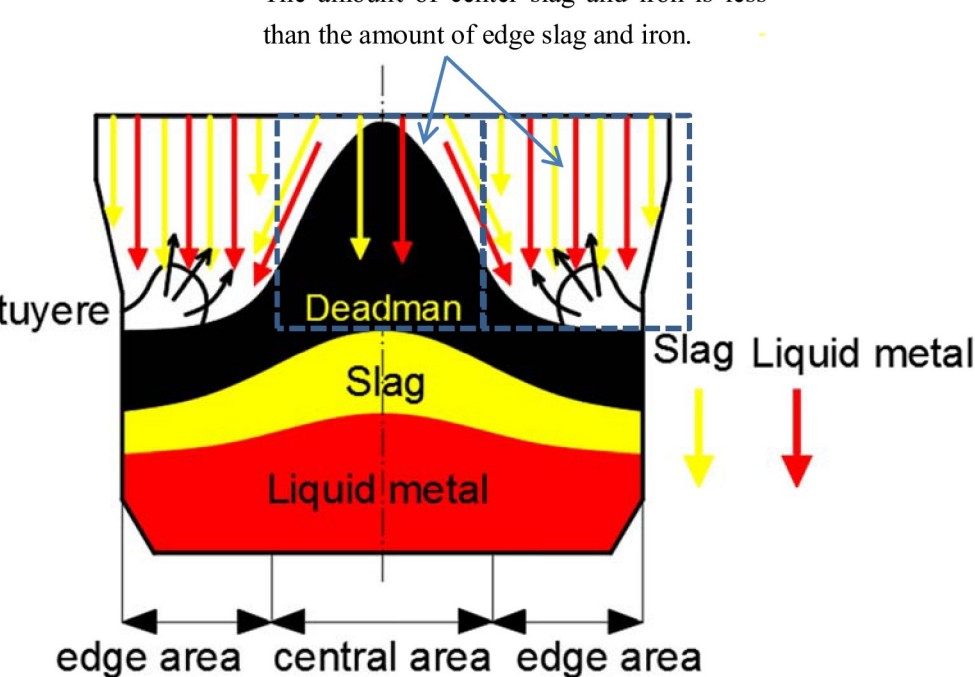

**Fig 4. Distribution of liquid metal and slag in the hearth of the 5,800 m³ blast furnace.**

detected inside the center coke at 2 m and 7.5 m from the hearth edge. Fig 5J shows that the slag is evenly distributed in the coke by filling 40–50% of the pores. Li [25] detected slag in coke obtained from the deadman and found that the slag contains akermanite ($Ca_2Mg$-$Si_2O_7$) and calcium aluminosilicate ($Ca_2Al_2SiO_7$). Subject to the high temperature and high pressure in the hearth, slag enters the coke through open pores. Fig 5C and 5D show that coke powder is in the slag, indicating that pore walls are eroded to some degree by the slag after it enters pores. Kasai [27] also proved that coke discharges coke particles when it reacts with slag. This is because slag may contain FeO, which may react with the coke surface. Fig 5H clearly illustrates that the slag flows in pores, erodes, and thins the pore walls, therefore lowering the coke strength.

### 3.3 Slag content in pores

Fig 6A shows that the coke is less than 10% slag at 0 m and 2 m from the hearth edge and more than 30% slag at 4 m, 6 m and 7.5 m from the hearth edge. This slag difference between edge coke and center coke can be explained in the following way. It takes time for the slag to enter the coke pores. Edge coke falls down quickly, and only a small amount of slag enters the coke surfaces; driven by the high content of hot liquid metal along the hearth edge, the reduction reaction and carburization effect assist the consumption of the coke surfaces containing the slag. Center coke, however, falls down slowly, which gives enough time for the slag to get deeper into it; coupled with the low content of liquid metal in the hearth center, the coke is barely eroded and consumed by the liquid metal.

Fig 6B shows that as the center coke descends to the bottom of the hearth, the amount of slag inside the pores increases from 46.5% to 53.6%. This is because the slag around the coke gradually enters the coke. The slag content in the coke pores at 5.2 m is substantially higher than that in the upper coke pores. This is because the position of 5.2 m is at the

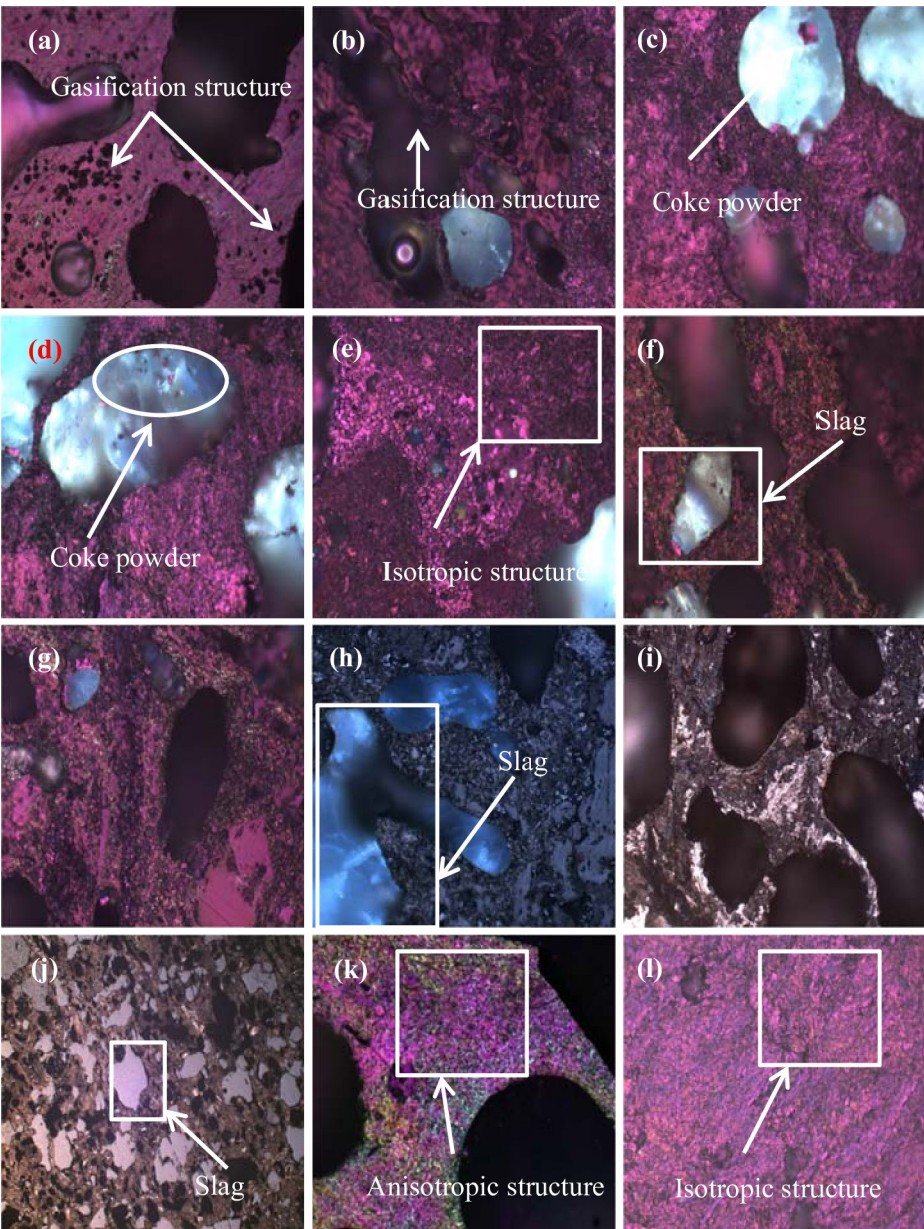

**Fig 5. Microstructures of the coke under the optical microscope.** (a) Sample ① under ×500 polarizing microscope with oil immersion lens; (b) Sample ② under ×500 polarizing microscope with oil immersion lens; (c) Sample ③ under ×500 microscope with oil immersion lens; (d) Sample ④ under ×500 polarizing microscope with oil immersion lens; (e) Sample ⑤ under ×500 polarizing microscope with oil immersion lens; (f) Sample ⑥ under ×500 polarizing microscope with oil immersion lens; (g) Sample ⑦ under ×500 polarizing microscope with oil immersion lens; (h) Sample ⑧ under ×500 polarizing microscope with oil immersion lens; (i) Sample ② under ×500 microscope with oil immersion lens; (j) Sample ⑤ under ×200 microscope with dry lens; (k) original coke under ×500 polarizing microscope with oil immersion lens; (l) ultragraphite morphology under ×500 polarizing microscope with oil immersion lens.

taphole level, where the slag content around the coke is higher than that in the upper part. The slag content in the pores increased substantially when the slag entered the pores at this position.

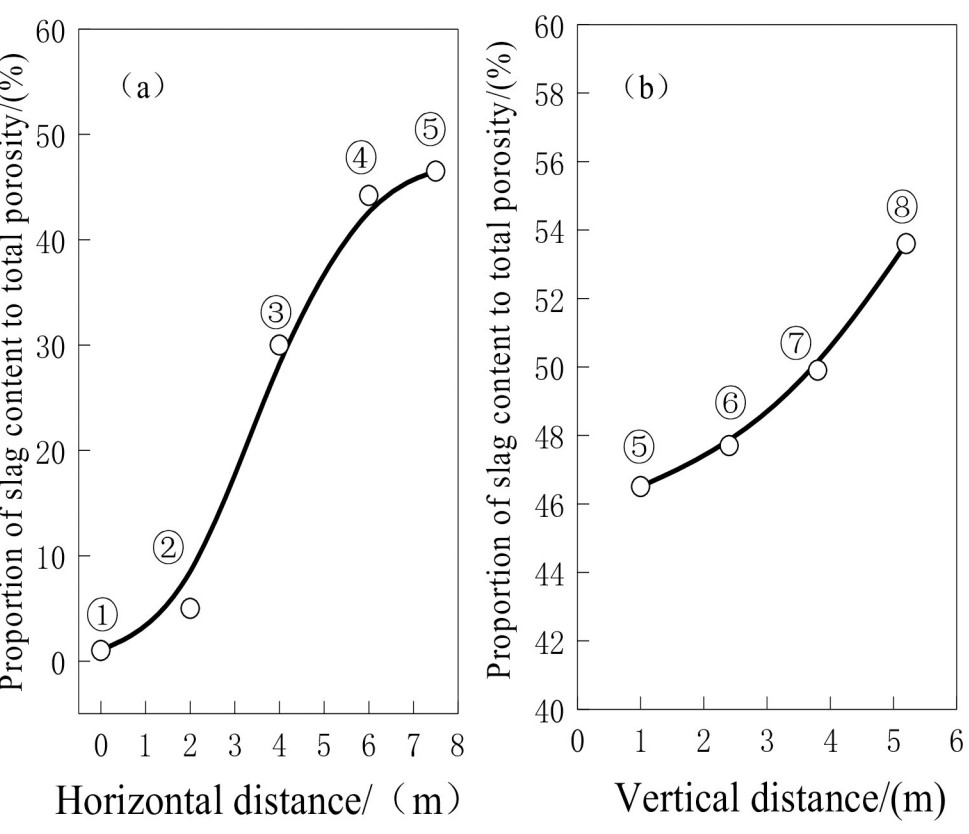

**Fig 6. Slag-to-porosity ratio.**

### 3.4 Graphitization comparison of the coke

Coke is produced through the pyrolysis of metallurgical coal and the polycondensation of pyrolysis products under high temperature in an oxygen-deficient environment. The essence of this coke-making process is that the side chains of aromatic nuclei that form the core part of the organic polymer are removed, and the resulting aromatic nuclei are polycondensed into ring structures, forming an organic solid with a graphite structure [28]. In the XRD patterns, this structure produces a peak at 26° that corresponds to the (002) lattice plane. According to the Scherrer equation, the average parameters of the stacking height can be calculated for microcrystalline graphite. The calculation method with detailed procedures was described in earlier publications [29].

$$L = K\lambda / (\beta\cos\theta)$$

The interlayer spacing, d002, can be calculated with Bragg's law:

$$d = \lambda / (2\sin\theta)$$

The average number of layers stacked per unit cell is

$$N = Lc/d$$

Studies [30, 31] indicate that after the coke is charged into the blast furnace, its degree of graphitization will increase. Figs 7A and 8A show that obvious peaks are observed at approximately 26° for coke. As shown in Fig 7B, the intensity of the (002) peak increases as the

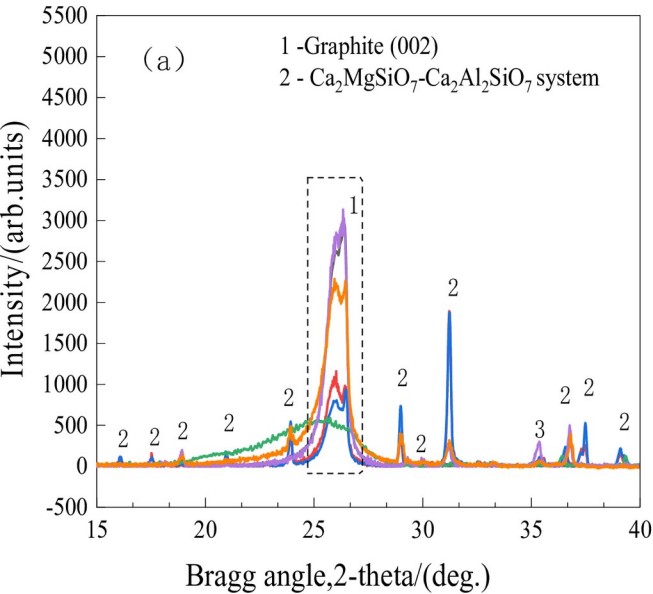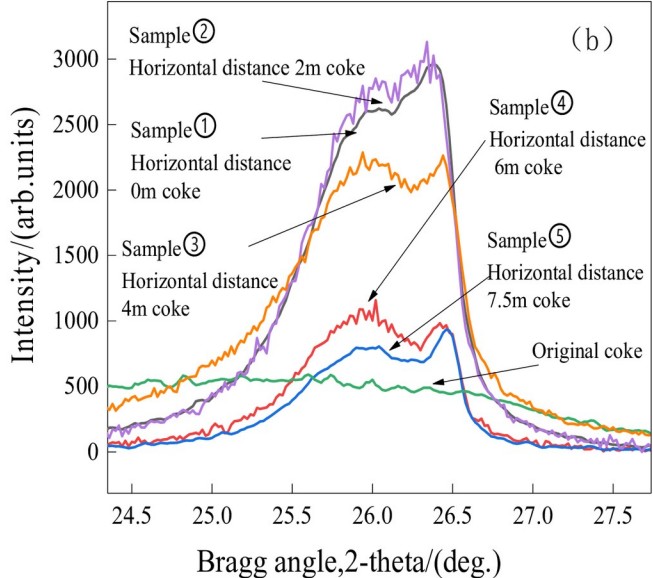

**Fig 7. XRD patterns of the coke vs. horizontal distance.**

sampling location moves toward the hearth center, indicating a decreased degree of graphitization. With the Scherrer equation and Bragg's law, the Lc values for the coke at 0 m, 2 m, 4 m 6 m, and 7.5 m from the hearth edge are 20.64 nm, 22.13 nm, 18.43 nm, 11.01 nm, and 10.45 nm, respectively, and the N values are 60, 65, 54, 32, and 30, respectively. For the original coke, the Lc is 8.69 nm, and N is 25. The coke in the hearth shows a remarkably higher degree of graphitization than the original coke because high temperature facilitates the transformation from coke to graphite. Fig 7B also shows that the graphitization degree of the coke at 0 m and 2 m from the hearth edge is higher than that at horizontal distances of 4 m, 6 m and 7.5 m. The hearth edge is near the tuyere zone, where much heat is produced by coke combustion; as a result, the degree of graphitization of the coke along the edge is higher than that of the coke in the center. The difference in the degree of graphitization of coke samples from 0 m and 2 m is minor and can be within the experimental error because of the strong mixing of coke particles within the raceway.

According to Fig 8B, the intensity of the (002) peak increases as the coke goes deeper into the hearth, indicating an increasing degree of graphitization; the Lc values for the coke at 1 m, 2.4 m, 3.8 m, and 5.2 m vertical distances are 10.45 nm, 10.62 nm, 10.69 nm, and 11.14 nm, respectively, and the N values are 30, 31, 32, and 33, respectively. This means that the coke undergoes a slowly increasing degree of graphitization as it goes toward the bottom of the hearth. Gupa [11] concluded that the degree of graphitization of coke powder is higher than that of bulk coke. In a study [32] with the tuyere sampling technique, researchers found that the coke subject to high temperature in the raceway zone is graphitized and that the degree of graphitization is higher on the surface of the coke than inside the coke, indicating that the graphitization process starts at the coke surface and goes inward gradually. According to the findings of Gupa [11], an increased degree of graphitization can lower the coke strength and result in its pulverization. It is implied that the degree of graphitization increases as the coke goes deeper toward the bottom of the hearth. Because the degree of graphitization is higher at the surface of the

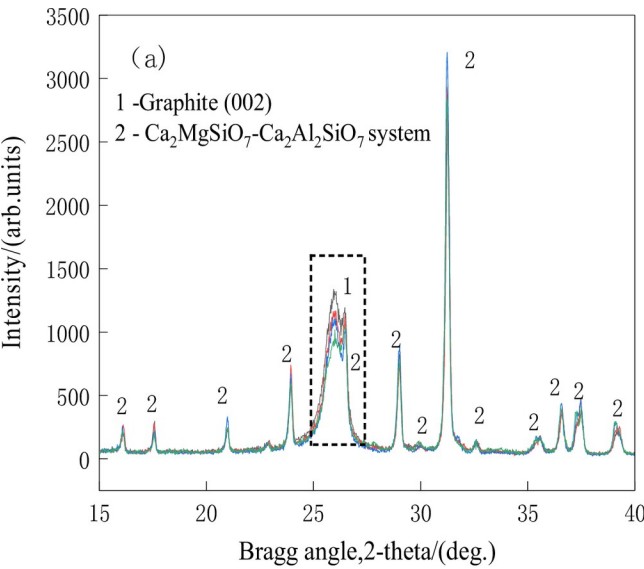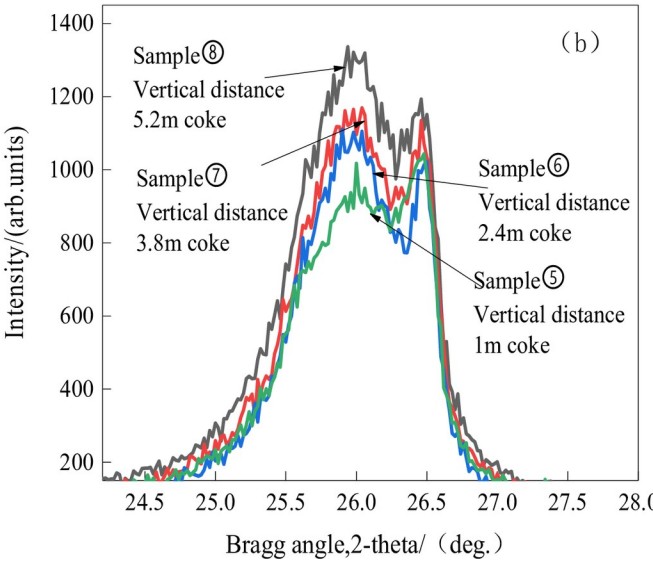

**Fig 8. XRD patterns of the coke vs. vertical distance.**

coke than in the core, under the pressure from furnace charge, the coke surface is easily removed, generating the coke powder.

As shown in Figs 7A and 8A, akermanite ($Ca_2MgSi_2O_7$) and calcium aluminosilicate ($Ca_2Al_2SiO_7$) diffraction peaks are found in the XRD patterns of the coke.

### 3.5 SEM analysis of the coke

**3.5.1 The process of slag uptake by coke.** Fig 9A and 9B, and (i) show that the coke at 0 m and 2 m from the hearth edge has coarser pore walls and larger pores than the original coke, which agrees with the result from Fig 5A and 5B. According to Fig 9C–9H, slag in different states is detected inside the coke at 4 m and 7.5 m from the hearth edge. Fig 9C demonstrates that the slag is dispersed within the 800 μm zone starting from the surface of the coke. Fig 9D shows that the slag that has just entered pores is droplet-like. Fig 9E shows that slag flows inside and fills pores. According to Fig 9H, large pores in the coke are substantially filled by slag. In previous studies [33, 34], researchers believed that the slag having gone into the coke would migrate to its surface as the coke is consumed, which would affect the dissolution of carbon from the coke to liquid metal. Researchers found that the effect of minerals on carburization is actually a speed control process under the influence of multiple factors [35]. According to a study, slag contained in coke will contact carbon bricks when the coke does so after it enters the liquid metal; in this case, calcium aluminates will react with alumina-silica refractory materials and cause the latter to expand, thereby adversely affecting the service life of carbon bricks [36]. According to the above analysis, the process by which slag enters coke easily occurs through open pores, and the dispersion of slag in coke through micropores is limited to the 800 μm zone starting from the coke surface. That is, by reducing the open porosity of the coke, we can substantially lower the total amount of slag that enters the coke, which can increase the carburizing rate and minimize the damage to carbon bricks.

The EDS analysis of the coke 2 m from the hearth edge is shown in Fig 10. The major element of the coke is C. The major elements of the ash contained in the coke are O, Al,

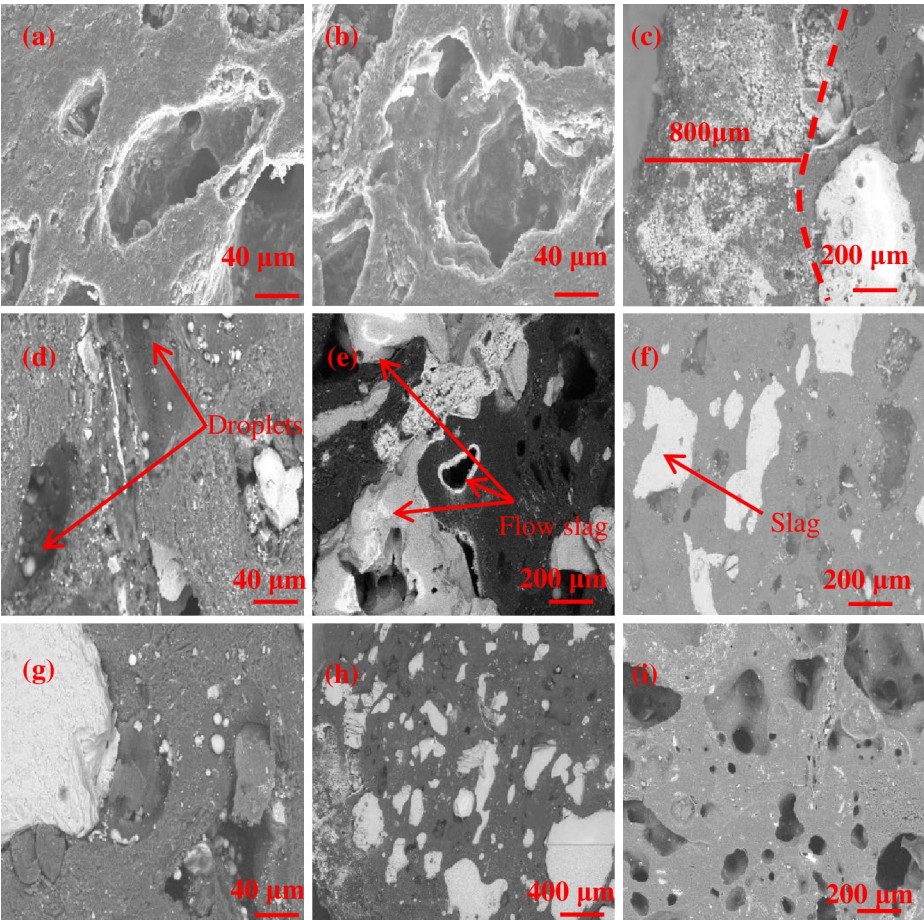

**Fig 9. SEM images of the coke.** (a) Sample ① at 0 m horizontal and 1 m vertical; (b) Sample ② at 2 m horizontal and 1 m vertical; (c) Sample ③ at 4 m horizontal and 1 m vertical; (d) Sample ④ at 6 m horizontal and 1 m vertical; (e) Sample ⑤ at 7.5 m horizontal and 1 m vertical; (f) Sample ⑥ at 7.5 m horizontal and 2.4 m vertical; (g) Sample ⑦ at 7.5 m horizontal and 3.8 m vertical; (h) Sample ⑧ at 7.5 m horizontal and 5.2 m vertical; and (i) original coke.

Ca, Mg and Si. The edge of the liquid metal is eroded. Iron, as shown in Fig 10 (P1), was found in the surface pores of coke. As indicated in Fig 3(i), the surface of the coke 2 m from the hearth edge is directly in contact with and eroded by the liquid metal. It is generally believed that coke plays a carburizing role for the liquid metal; it is also found from Fig 10 that molecules in the liquid metal can penetrate the coke through surfaces on which they contact the coke. The area-scan images of the coke at the 7.5 m horizontal and 5.2 m vertical position under EDS are shown in Fig 11. The major elements of slag are Ca, Si, Al, Mg, and O, a result that is consistent with the XRD patterns. Chang [23] found that slag is detected in coke pores in the cohesive zone of the blast furnace. The major elements of slag are also Ca, Si, Al, Mg, and O. It can be inferred that the process of the slag entering the coke begins from the cohesive zone of the blast furnace. With the renewal of the deadman, the slag inside the coke descends to the hearth of the blast furnace together with the coke. Interestingly, the Si content is higher at the edge of the slag in the pores than in the interior. This finding may be due to the formation of coke ash after coke gasification. The Si content is substantially higher in the coke than in the blast furnace slag. The ash content in the surface layer of coke pores is considered a hinderance to coke

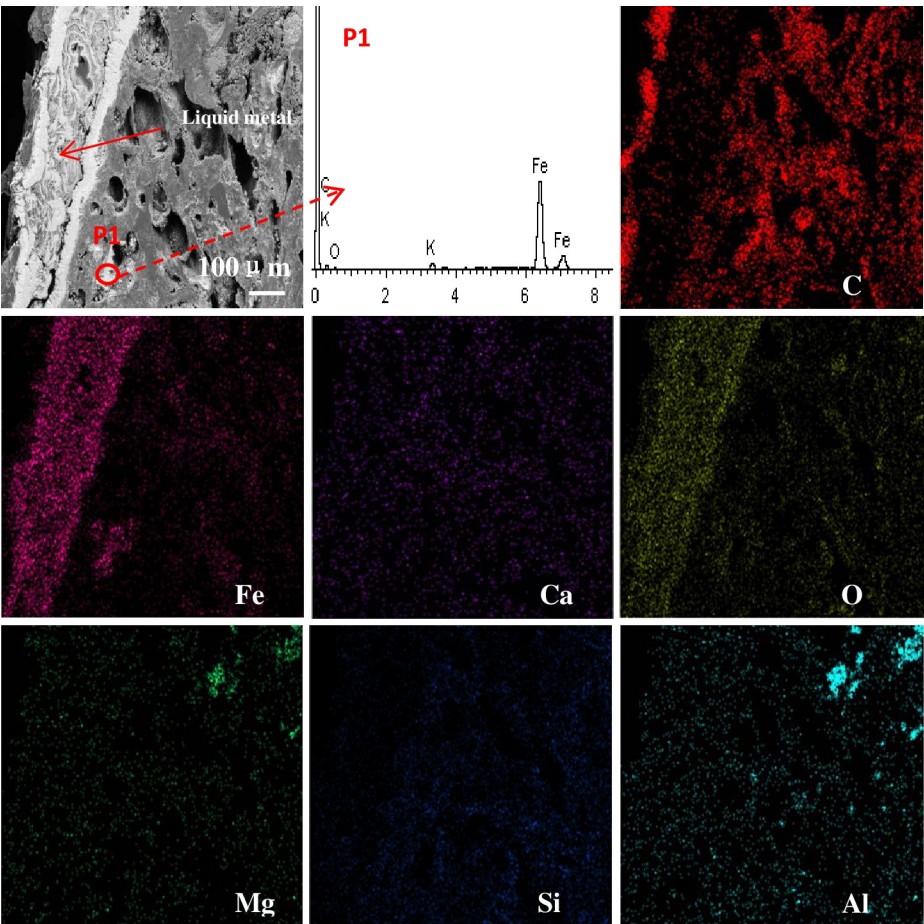

**Fig 10. EDS images of the coke at 2 m horizontal and 1 m vertical.**

gasification and slag erosion [25]. At the same time, a small amount of Fe was found at the edge of the slag, which may be FeO.

**3.5.2 Deterioration mechanism of the coke in the hearth.** As shown in Fig 12A–12C, the coke 2 m from the hearth edge has smooth surfaces. Coke edges have been eroded into round shapes, with open pores detected. Coke surfaces are eroded because of the carburizing effect enhanced by the high flow rate of liquid metal. Fig 13A shows the process in which the edge coke decreases because of the carburizing effect at the 2 m horizontal and 1 m vertical position. Fig 13B provides a graphic illustration of this process.

Fig 14A shows that slag penetrates into gaps between the coke crystals. According to Fig 14B, many flakes with a crystalline graphite-like structure are on the coke surface [37–40], indicating the high degree of graphitization of the coke surface, and cracks are also found in these flakes. Fig 14C shows the cracks detected on the pore walls of the coke. Many low-strength crystalline flakes are formed on the coke surface because of the graphitization process, and such flakes are easily pulverized under pressure from the furnace charge. The high pressure from the blast furnace blower inserts the slag into gaps between these flakes and causes cracks on the internal pore walls, thereby further lowering the strength of the coke. Fig 15A illustrates the pulverization process of coke in the hearth

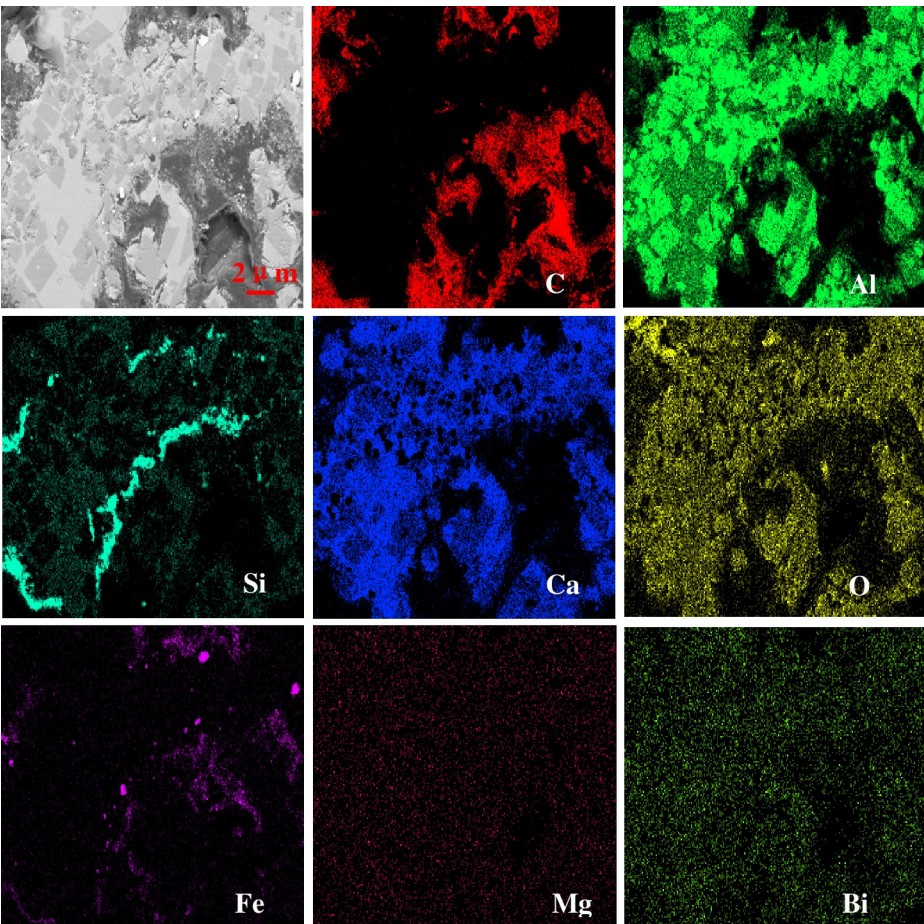

**Fig 11. EDS images of the coke at 7.5 m horizontal and 5.2 m vertical.**

center at the 7.5 m horizontal and -3.8 m vertical position. Fig 15B provides a graphic illustration of this process.

The above discussion indicates that because of the uneven distribution of molten iron in the superlarge blast furnace, different mechanisms for coke particle size reduction are undertaken. The edge coke decreases because of the carburizing effect. Slag inserted into micropores on the surface of the center coke loosens the surface structure of the coke, which, coupled with

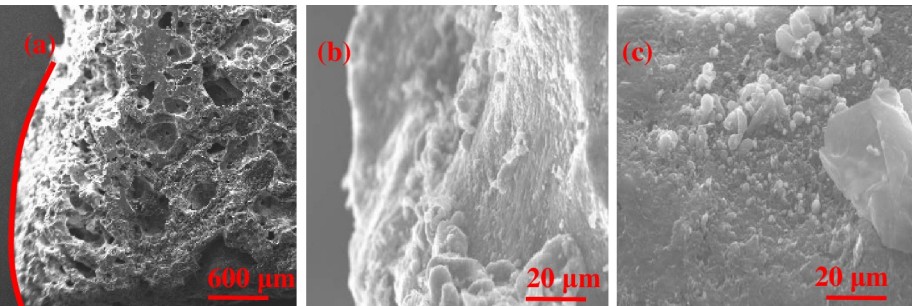

**Fig 12. Surface microstructure of edge coke at 2 m horizontal and -1 m vertical.**

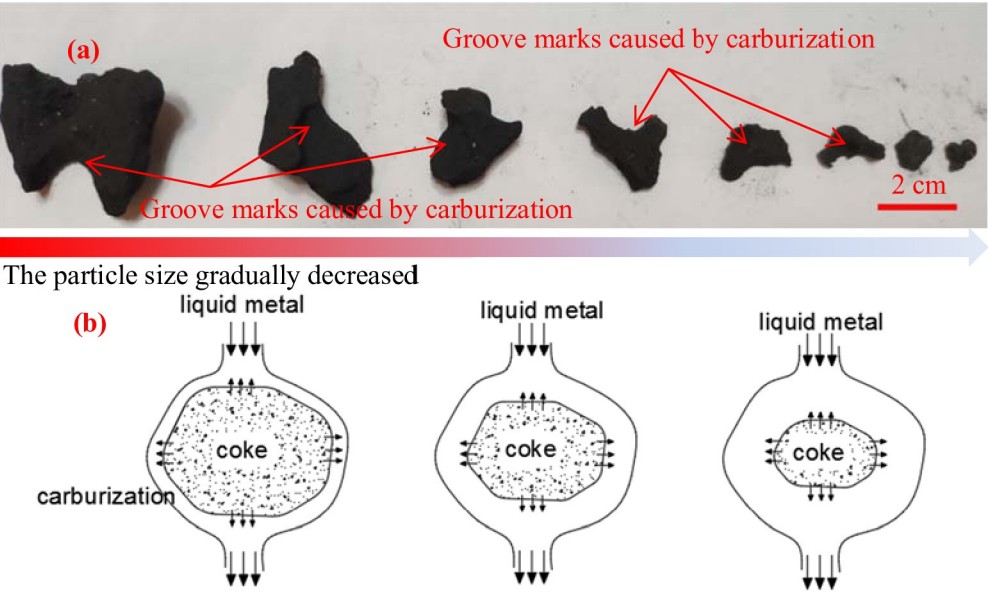

**Fig 13. Disappearance mechanism of edge coke.**

the flake graphite-like structure appearing on the surface of the coke, lowers the strength and causes cracks and consequent degradation of the coke.

## 4. Conclusions

The edge coke and center coke in the 5,800 $m^3$ blast furnace differ significantly from each other regarding macromorphology, degree of graphitization, microstructure, and deterioration mechanism because of the uneven distribution of the coke in the blast furnace.

Center coke is basically free from erosion, whereas edge coke is severely eroded by liquid metal with the production of groove marks. Compared with the original coke, the coke in the hearth shows a remarkably higher degree of graphitization and isotropic structure instead of an anisotropic structure. Edge coke undergoes a higher degree of graphitization than center coke. Center coke undergoes an increasing degree of graphitization as it descends to the bottom. The deterioration mechanism differs between edge coke and center coke. The surface structure of center coke loosens upon the insertion of slag into the micropores on the surface of the coke, which, coupled with the flake graphite-like structures appearing on the surface of

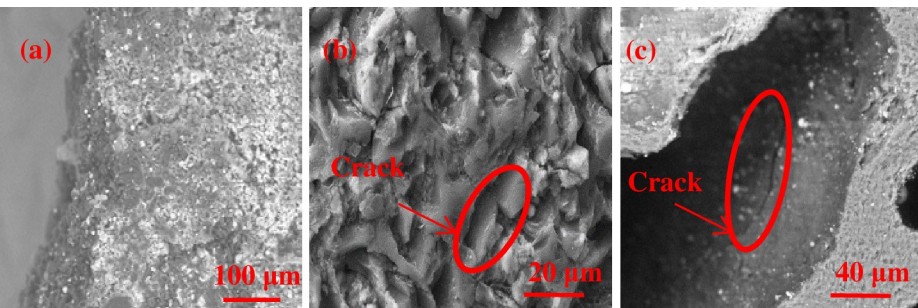

**Fig 14. Surface microstructure of center coke at the 7.5 m horizontal and 1 m vertical position.**

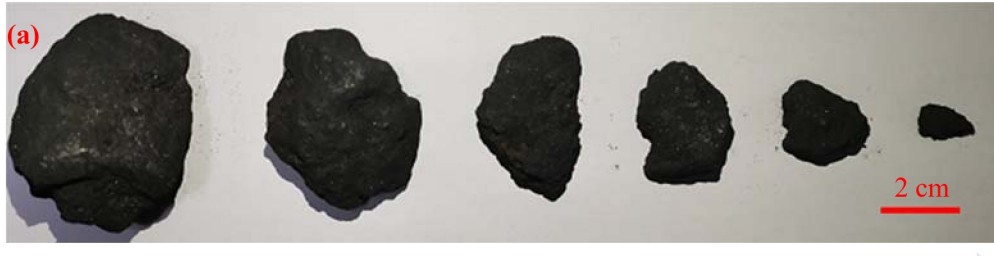

The particle size gradually decreased

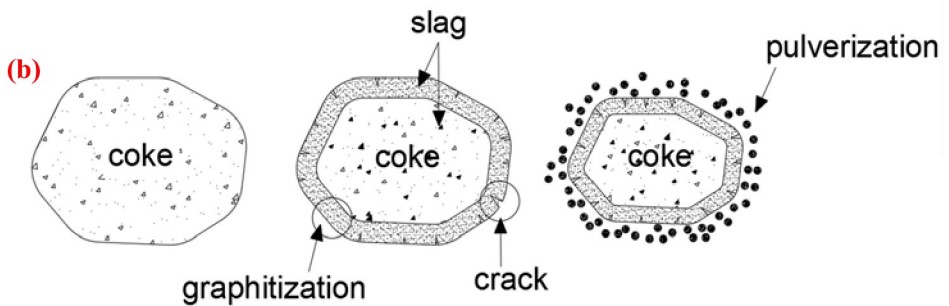

**Fig 15. Pulverization of center coke.**

the coke, lowers the strength and causes cracks and consequent degradation of the coke. The edge coke decreases and eventually disappears due to the carburizing effect.

## Supporting information

**S1 Fig. The slag obtained from the 5,800 $m^3$ blast furnace.**
(TIF)

**S2 Fig. SEM and EDS analysis.** The results of an energy spectrum analysis show that the main components of blast furnace slag are Ca, Al, Mg, Si, and O, which is consistent with the results in the references. A small amount of S was also found in the slag. S comes from coke, and sulfur is absorbed by blast furnace slag.
(TIF)

**S1 File.**
(DOCX)

## Author Contributions

**Conceptualization:** Ping Du.

**Formal analysis:** Yong S. Tian.

**Investigation:** Jun L. Zhou.

**Writing – original draft:** Qing Q. Lv.

**Writing – review & editing:** Qing Q. Lv, Guang H. Wang.

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
