## [Decision Letter · Decision Letter 0]

9 Nov 2020

PONE-D-20-30004

A Study on the Characteristics of Coke in the Hearth of a Super-large Blast Furnace

PLOS ONE

Dear Dr. Lv,

Thank you for submitting your manuscript to PLOS ONE. After careful consideration, we feel that it has merit but does not fully meet PLOS ONE’s publication criteria as it currently stands. Therefore, we invite you to submit a revised version of the manuscript that addresses the points raised during the review process.

We look forward to receiving your revised manuscript.

Kind regards,

Antonio Riveiro Rodríguez, PhD

Academic Editor

PLOS ONE

2.Thank you for stating the following in the Financial Disclosure section:

[Fund name:Supported by National Key R&D Program of China

Fund number:(2017YFB0304300&2017YFB0304303)

Author Name:wang guang-hui

The funders had no role in study design, data collection and analysis, decision to publish, or preparation of the manuscript.].   

We note that one or more of the authors are employed by a commercial company: Jiangsu Huade Hydrogen Energy Technology Co., Ltd.,

3. PLOS specifies that experiments, statistics, and other analyses are performed to a high technical standard; sample sizes are large enough to produce robust results; and methods are described in sufficient detail to allow another researcher to reproduce the experiment (http://journals.plos.org/plosone/s/criteria-for-publication#loc-3). As such, we ask you to revise your Methods section to include more detailed information, such that a reader could replicate the findings of your study. For example, you do not provide the name, model, or manufacturer of the SEM or its accessories. We would expect this information for all instruments and/or chemicals used in the study.

Reviewers' comments:

Reviewer's Responses to Questions

**Comments to the Author**

1. Is the manuscript technically sound, and do the data support the conclusions?

Reviewer #1: Yes

Reviewer #2: Yes

2. Has the statistical analysis been performed appropriately and rigorously? 

Reviewer #1: N/A

Reviewer #2: No

3. Have the authors made all data underlying the findings in their manuscript fully available?

Reviewer #1: Yes

Reviewer #2: Yes

4. Is the manuscript presented in an intelligible fashion and written in standard English?

Reviewer #1: No

Reviewer #2: Yes

5. Review Comments to the Author

Reviewer #1: The manuscript reports valuable data on the characteristics of the hearth coke samples from a very large blast furnace. This type of data are rare to obtain in the literature.

A major drawback of the work is that the manuscript focuses too much on the coke samples themselves without linking their characteristics to the reactions in a blast furnace, that is the effects of gasification and combustion in the raceway.

Figs. 15 and 17 are not found in the submission.

Figure 1, the current marking of distances from the tuyere level should be positive values, not negative. The marking of distances 5.2 m and 4.4 m external of the furnace hearth section is unnecessary.

Figure 4, schematic demonstration

Figure 6 shows the change in mean particle size with location. How is the mean particle size calculated? The authors state that “The extremely high content of the coke powder in the hearth center significantly decreases the arithmetic mean particle size of the coke in this area.” In the view of the reviewer, the fine particles with size <10 mm should be excluded from the calculation to improve the certainty (that is, what value is used for the fine particles with particle size <10 mm?).

The authors state that “As shown in Fig.9(b), the intensity of (002) peak increases as the sampling location moves toward the hearth center, an indicator of the increased degree of graphitization.” “Increases” and “increased” should be decreased here to match the results.

The authors discussed the difference in the degree of graphitisation of coke samples from 0 m and 2 m. This discuss may not be sensible because of a strong mixing of coke particles within raceway. The difference is minor and can be within experimental error.

Figures 9 and 10: The labelling of mineral phase Ca2MgSi2O7 is incorrect.

In relation to the discussion in section 3.5, separate XRD analyses of the coarse and fine coke particles and differentiation of their degree of graphitisation help understanding of the source of fine coke particles, especially for samples 5 to 8.

With regards to the discussion on Fig. 11 by SEM analysis, it is not clear how the images are representative of a whole coke sample, and whether the authors tried to distinguish between the coke ash and the intruded bulk slag within coke particles.

On Fig. 12, “major elements of the coke are O, Al, Ca, Fe, and Si”?

Current observation presented in Fig. 12 does not convincingly connect the presented Fe with metal phase. It is necessary to magnify the zone and determine the composition of the related phase by EDS to confirm that a metallic phase is present, rather than FeO in the slag.

Reviewer #2: This manuscript investigated the mechanism of coke degradation in the blast furnace caused by its interaction with liquid iron and slag. This topic is interesting and importance for the steelmaking industry. However, major revision is required before publishing this manuscript as a research paper in PLOS ONE. The detailed comments are shown as below:

1. The introduction of this manuscript needs to be significantly improved. It needs to cover an introduction of research background, current development of this area, what have been done in the previous studies, what have not been fully investigated previously, and the importance of the investigation in this manuscript. The current one paragraph introduction is far from enough.

2. Section 2.2. This manuscript determined the slag/pore ratio by calculating the ratio between their number of spots. Are the number of spots determined manually or by some software? Should this ratio be calculated by their area ratio? Please provide more details about this measurement.

3. Section 3.2. Please provide the statistic analysis and error analysis of the particle size distribution measurement.

4. Section 3.3. Authors were trying to analysis the slag composition of cokes extracted from BF. However, such analysis was only based on the literature review. Do you have any experimental data, such as EDS, to support your argument?

5. Section 3.4. Same comment as comment 2. If this measurement was based on the ratio of number of points between slag and pore. The discussion in this section was not very convincing. An ideal measurement is the area ratio.

6. PLOS authors have the option to publish the peer review history of their article (what does this mean?). If published, this will include your full peer review and any attached files.

Reviewer #1: No

Reviewer #2: No

---

## [Author Response · Author response to Decision Letter 0]

3 Jan 2021

Dear two reviewers, 

It is a great honor to receive the comments on this manuscript. Thank you for your patience. All the comments are very professional. I learned a lot from it. I have revised the manuscript to meet the requirements and the format of PLOS ONE. And I have chosen AJE for language editing of my manuscript. I hope that the resubmitted manuscript can meet the requirements of PLOS ONE.

If the manuscript still needs to be improved, please give feedback in time, and I will make corresponding modifications.

Thank you very much.

---

## [Decision Letter · Decision Letter 1]

1 Feb 2021

A Study on the Characteristics of Coke in the Hearth of a Superlarge Blast Furnace

PONE-D-20-30004R1

Dear Dr. Lv,

We’re pleased to inform you that your manuscript has been judged scientifically suitable for publication and will be formally accepted for publication once it meets all outstanding technical requirements.

Kind regards,

Antonio Riveiro Rodríguez, PhD

Academic Editor

PLOS ONE

Reviewers' comments:

Reviewer's Responses to Questions

**Comments to the Author**

1. If the authors have adequately addressed your comments raised in a previous round of review and you feel that this manuscript is now acceptable for publication, you may indicate that here to bypass the “Comments to the Author” section, enter your conflict of interest statement in the “Confidential to Editor” section, and submit your "Accept" recommendation.

Reviewer #2: All comments have been addressed

2. Is the manuscript technically sound, and do the data support the conclusions?

Reviewer #2: Yes

3. Has the statistical analysis been performed appropriately and rigorously? 

Reviewer #2: No

4. Have the authors made all data underlying the findings in their manuscript fully available?

Reviewer #2: Yes

5. Is the manuscript presented in an intelligible fashion and written in standard English?

Reviewer #2: Yes

6. Review Comments to the Author

Reviewer #2: This manuscript conducted an investigation about the coke behaviour in the large blast furnace, which was valuable for both industry and researchers in the same area. This revision version has addressed comments from the reviewers. The introduction session has been improved significantly compare to the original submission, and the analysis of the experimental data has been detailed as well. Therefore, I recommend this manuscript to be accepted in Plos One as a research paper.

7. PLOS authors have the option to publish the peer review history of their article (what does this mean?). If published, this will include your full peer review and any attached files.

Reviewer #2: No

---

## [Editor Report · Acceptance letter]

19 Feb 2021

PONE-D-20-30004R1 

A Study on the Characteristics of Coke in the Hearth of a Superlarge Blast Furnace 

Dear Dr. Lv:

I'm pleased to inform you that your manuscript has been deemed suitable for publication in PLOS ONE. Congratulations! Your manuscript is now with our production department. 

Kind regards, 

on behalf of

Dr. Antonio Riveiro Rodríguez 

Academic Editor

PLOS ONE